# Influence of Long-Period-Stacking Ordered Structure on the Damping Capacities and Mechanical Properties of Mg-Zn-Y-Mn As-Cast Alloys

**DOI:** 10.3390/ma13204654

**Published:** 2020-10-19

**Authors:** Ruopeng Lu, Kai Jiao, Yuhong Zhao, Kun Li, Keyu Yao, Hua Hou

**Affiliations:** College of Materials Science and Engineering, North University of China, Taiyuan 030051, China; jiaokai0608@hotmail.com (K.J.); likun1027@hotmail.com (K.L.); qmnmna@outlook.com (K.Y.); houhua@nuc.edu.cn (H.H.)

**Keywords:** magnesium alloys, Mg-Zn-Y-Mn alloys, damping, long-period stacking ordered structure

## Abstract

Magnesium alloys are concerned for its mechanical properties and high damping performance. The influence of Mn toward the internal organization morphology of long-period stacking ordered (LPSO) second phase and the consistent damping performance in Mg-4.9Zn-8.9Y-xMn have been studies in this work. It has shown that the addition of Mn tends to diffuse to the LPSO interface and causes the LPSO phase to expand in the arc direction. The circular structure of LPSO can optimize the damping property of the alloy better than the structure with strong orientation, especially at the strain of 10^−3^ and 250 °C. With more additions of Mn, damping would have a reduction due to the dispersed fine LPSO phases and α-Mn particles. When the Mn content is higher than 1.02%, the grain is refined, and mechanical properties have been significantly improved. Mg-4.9%Zn-8.9%Y-1.33%Mn shows the best mechanical property.

## 1. Introduction

Nowadays, with the rapid development of society and modern industries, such as aerospace [1,2], weaponry, and transportation, the problems that have been induced by vibration and noise have become much more important [3]. Under this condition, the current metal-based damping alloy is not only a structural material [4], but also a functional material because it can reduce vibration and noise, so it has attracted much attention in industrial applications [5,6].

The damping of the material is also called internal friction. It is not necessary to add damping components and only consume energy through itself [7]. Studies have shown that high-purity magnesium has very good damping performance [8,9], which is much higher than other metal materials. However, the addition of most alloying elements will reduce its damping characteristics [10,11,12]. Presently, a series of damping alloys have been used in industry or prepared in the laboratory, such as Mg-Zr, Mg-Mn, Mg-Ni, and Mg-Cu-Mn, but the mechanics of these damping magnesium alloys are still not good enough [13,14,15], even when compared with common magnesium alloys. Therefore, how to improve the mechanics of damping magnesium alloys or effectively improve the damping characteristics of common magnesium alloys is a scientific question worthy of study [16].

According to previous studies [17], it is shown that, along with the increase of LPSO volume percentage, both the damping and mechanical performance are improved. The LPSO phase that is produced during solidification has a unique role in the damping and mechanics of magnesium alloys, and it can improve both properties at the same time. As the most common ternary alloy with LPSO phase, Mg-Zn-Y series has attracted many attentions in damping alloys [18]. Some work indicate that the LPSO second phase in the alloy are able to present a variety of morphological characteristics in Mg-Zn-Y ternary series magnesium alloys; the formation of rod-shaped structure can improve the damping characteristics of the alloy and also has a similar fiber-reinforced effect on mechanics [19,20].

However, direct cooling to obtain as-cast Mg-Zn-Y alloys has the comparatively large grain dimension with a certain number of impurities. Usually, Zr is an excellent element in refining the metallographic microstructure meanwhile improving mechanical properties, and Mg-Zn-Y-Zr quaternary alloys also show better damping capacity [21,22]. As another grain refining element in Mg alloys, the addition of Mn can adsorb Fe and other magazine elements in order to clean the melt. Moreover, an appropriate Mn element is an effective element in Mg alloys with a lot of dispersed phases which can cause in long dislocation introduce to the alloys, promoting the dislocation slipping and improving their damping capacity [23,24,25].

As above, it provides new research ideas for design high damping alloys, which is to add Mn element and appropriate heat treatment process to control the alloy morphology. In this study, the influence of the addition of different content of Mn element on the damping of Mg-Zn-Y alloy was discussed.

## 2. Materials and Methods

Pure Mg, pure Zn, Mg-30wt%Y, and Mg-3%Mn (Regal-metal, Shanxi, China) were used as raw materials in order to obtain the Mg-Zn-Y-Mn (Table 1) alloys. All of the master materials were mixed and melted in an electrical-magnetic furnace while using a mild stainless steel crucible under a protective argon gas environment of 820 °C. During the melting, the metal liquid was stirred by an electromagnetic field. After being stirred uniformly, the melting liquid was standing and holding the temperature for 15 min. Then the molten metal was poured into the metal mold placed in water, thus obtaining the metal ingot. The nominal and actual chemical compositions were list in Table 1.

The metallographic microscope was observed by optical microscope (OM, Leica Microsystems, Marseille, France). The internal organization was observed after further magnification by scanning electron microscope. The equipment is Vega II LMU scanning electron microscope (SEM, Vega II, Brno, Czech Republic). The phase composition of different parts of the interior was analyzed by energy-dispersive spectrometer (EDS, Vega II, Brno, Czech Republic). Use X-ray diffractometer (XRD, Rigaku D/MAX2500PC, Tokyo, Japan) and standard PDF card to identify and analyze the phase, the radiation is carried out in the range of 20°–90° at a speed of 1.5°/min. The XRD results were phase calibration by using the Jade 6.0 software (Philips, New Zealand, Christchurch).

A dynamic mechanical analyzer (TA-DMA Q800, Chicago, USA) was used in order to test the damping performance in single-cantilever vibration mode. The damping performance is evaluated using formula Q^−1^ = tanθ; here, θ is defined as the hysteresis angle between the added strain and the corresponding stress. Q^−1^ is the inverse of the quality factor, used to measure the capacities of the material damping. A damping sheet test sample with a size of 45 mm × 5 mm × 1.2 mm was prepared with a wire-cut electric discharge machine. To obtain the dependence of damping performance on strain, test on strain amplifiers in the range of 1 × 10^−5^ to 1 × 10^−3^ at room temperature and the vibration frequency (*f* = 1 Hz). The room temperature tensile testing machine (Shimadzu CMT-5105, Tokyo, Japan) at the stable tensile speed of 3 mm/min. and the primary strain speed of 1.2 × 10^−3^ s^−1^.

## 3. Results and Discussion

### 3.1. Microstructure of the As-Cast Mg-Zn-Y-Mn Alloys

Figure 1 shows the OM images of the as-cast Mg-Zn-Y-Mn alloys. The figures show that the alloy metallography contains two parts of the structure, one is equiaxed dendrites of α-Mg (Yellow dotted line) and the other is the second phase structure at the grain boundary. The average grain dimension of alloy I is about 20 μm. The Mg-4.9Zn-8.9Y alloy without Mn addition contains a large amount of dendrites, as indicated by the yellow dotted line in Figure 1a. Figure 1b shows that the grains and dendrites became finer, and some particles are presented in alloy II. It also shows that the grains gradually changed from dendrites to equiaxed grains (the area indicated by the red arrow). The particles gathered to form a cluster structure in alloy III (the red circle in Figure 1c,d). While Mn reached 1.02% in alloy IV, the grain of the alloy is more refined to 12 μm. At this time, it was found that the orientation of the phase growth weakened described by the yellow circle, which may be caused by the divorced growth of the phase. While Mn reached 1.37% in alloy V, the second phase increases significantly (the red circle in Figure 1e); meanwhile, the divorced growth pattern that is related to the increase of Mn content is more significant (yellow dotted line).

Figure 2 shows the XRD patterns of these Mg-Zn-Y alloys with different Mn content. The result shows that all of the alloys are mainly constituted of α-Mg and Mg_12_YZn (LPSO) phases two parts. The LPSO phase has a strong diffraction peak at 2θ of about 32°, 35°, and 40°.

Figure 3 shows the SEM images of the alloys, it can clearly see the dendrite structure and second phase. The secondary phase presents a continuously distributed gray body on the grain boundary and forms a network structure. Some bright white particles are dispersed in the matrix. Figure 3a shows that the second phase of the alloy without Mn has an obvious orientation, the layered LPSO in each grain is parallel to each other. Figure 3b,c show that, with the addition of Mn, the orientation of LPSO arranged at the grain boundary decreases. Interestingly, the main LPSO phases do not grow parallel to each other, and their morphology expands toward the arc of the yellow arrow. In Figure 3d, the preferential growth of LPSO is no longer obvious, and the secondary phase expands and connects to form a network along the arc direction, as shown in the yellow dotted line. This phenomenon is consistent with the speculated results of the metallographic observation. While Mn reached 1.37% in alloy V, it is observed that bright white dots are surrounded by the LPSO phase in Figure 3e. With the addition of Mn, the as-cast morphology of the alloy changed slightly.

Figure 4 shows the structure morphology observed by SEM and EDS element spectrum of the as-cast alloy V. Table 2 shows the EDS elemental analysis of the as-cast Alloy V specified in Figure 4. The gray net-shaped bulk phase, as the main secondary phase, is observed with RE/Zn of 3/2 in point A, E, and F respectively. According to the XRD patterns in Figure 1, it could be suggested as an LPSO phase. The bright particles display two main morphologies. The snowflakes-shaped particles contain a lot of Mn, which attached to the gray LPSO in Figure 4a. The solute atoms will deform the crystal lattice, change the lattice parameters, and form clusters with surrounding atoms. Because of the diversity of electronegativity, a variety of atomic clusters will be produced [9]. Based on the EDS results in Table 2, the particles could be inferred as α-Mn particles or Y-rich particles. It could be concluded that α-Mn particles are a loose structure and they often accumulate at the grain boundary, as marked in point B. However, Y-rich particles usually appear in the matrix with regular shape and almost do not gather into large pieces, marked as C. Some dark phase (point D) in the alloys are determined to be α-Mg matrix. Figure 4d,e show the distribution of element of SEM microstructure in alloy V, alloy II respectively. The existence of Mg is mainly Mg-Zn-Y phase. It shows when the Mn content is added to 1.37%, the Mg_12_YZn phase transition becomes more uniform due to the divorced growth leading to the transformation of the LPSO phase morphology. There is a clear trend of aggregation for the distribution of Mn.

Figure 5 is a simplified diagram of the effect of Mn on the microstructure of Mg-4.9Zn-8.9Y series alloys. Figure 5a,b show the schematic diagram of divorced growth of Mg-4.9Zn-8.9Y alloy without Mn addition. When the solidification α-Mg is basically completed, the second phase is first formed in the remaining liquid phase, then Y particles. Figure 5c,d show that, after the addition of the Mn element, the Mn element tends to diffuse to the LPSO interface, which hinders the LPSO orientation growth and causes the LPSO phase to expand in the arc direction. The LPSO phase also drives out the surrounding Mn elements during the growth process. When the Mn content is added enough, part of the Mn atoms gather at the grain boundaries and Y atoms are wrapped in the LPSO second phase, as shown in Figure 5e, and the actual situation is just like Figure 4d,e. This growth mode describes that the Mn element can change the shape of the LPSO structure and improve its growth orientation.

### 3.2. Damping Performance of the Mg-Zn-Y-Mn Alloys

Figure 6 shows the damping performance of the Mg-Zn-Y-Mn series alloys at room temperature. Magnesium damping is generally regarded as a dislocation-type damping mechanism, which conforms to the Granato-Lücke theory [26,27]. When the strain is low, the damping has nothing to do with the strain, and the mechanical energy will be consumed by the bowing motion of the dislocation line in the alloy. When the strain outrides the critical place, the damping performance of the alloy is greatly increased. This state is regarded as the dislocation line breaking free from the bondage of the weak pinning point and dragging the pinning point to move together. For complex systems, the critical unpinning strain is often not obvious. In this experiment, it can be observed that, when the strain reaches 10^−4^~2*10^−4^, the damping begins to increase significantly. By deeply studied, all of the damping capacities (Q^−1^) of as-cast alloys reached 0.02 at the critical strain of 10^−3^, meaning high damping alloys. Among these, alloy III shows the better damping capacities on the low strain area (when the strain is 10^−4^) with the Q^−1^ value of 0.003, while no obvious difference happened on the other four alloys. On high strain area (when the strain is 10^−3^), with the increment in strain, Q^−1^ values of the five alloys show clearly different and show the trend where alloy II, alloy III, and alloy IV exhibited slightly higher damping values than the other two. Especially in the high strain stage, this performance is more obvious.

Figure 7 shows the damping performance of these five alloys at strain 10^−3^. Alloy II with 0.44% Mn shows the best damping performance. There are two reasons for this phenomenon, first, the addition of a small amount of Mn causes a large number of dislocations around the Mn to appear due to the different thermal expansion coefficients during the alloy cooling process, which provides movable dislocations for the alloy, which is beneficial for improving the damping; on the other hand, the Mn content is low, at 0.44%, the anisotropy of the second phase is more obvious, the LPSO morphology has not been completely rounded, and the energy absorption effect on the alloy is not good. Combined with the schematic diagram of OM and SEM organization changes, it can be known that the Mn element increases the divorced growth pattern and makes the second phase shows a more pronounced change in the arcing trend results from the improved energy absorption of the alloy. Small amounts of Mn can change the shape of the LPSO phase and improve the damping capacities in Mg-Zn-Y alloys. Therefore, when Mn is 0.44%, the increase in damping comes from the change of the second phase morphology and dislocation content, and these two factors are closely related to the Mn element.

According to the previous studies of Mg-Mn binary alloys [14], appropriate Mn particles that dispersed uniformly in α-Mg matrix could result in longer dislocation and improving the damping capacity, but excessive Mn element will cause the mutual blocking of dislocations, thereby reducing damping. It is precisely because of such a complicated situation that, as the Mn content increases, the damping of alloys III and IV is indistinguishable from that of alloy II, but it is still higher than that of alloy I. The damping performance of the alloy is struggling in three factors: the LPSO phase morphology, increase of dislocations, and pinning of dislocations.

With more Mn, the damping performance of alloy V is significantly reduced. Combined with the microstructure, it can be inferred damping values would have a reduction due to the dispersed fine LPSO phases and excessive α-Mn particles. On the contrary, it will promote the morphological transformation of the LPSO phase, reduce the anisotropy of the second phase, and improve the damping performance.

Figure 8 shows the Q^−1^ of the Mg-4.9Zn-8.9Y-1.33Mn alloy and the other comparatively protruding damping materials, such as Mg-Ni alloy [28], AZ63 alloy [29], Mg-8Li-Al alloy [30], CM31 alloy [31], and so on. The current Mg-4.9Zn-8.9Y-1.33Mn alloy exhibits an excellent overall performance with Q^−1^ of 0.02, which almost reached the level of high damping alloy Q^−1^ of 0.01.

The temperature-dependent damping capacities of Mg-Zn-Y-Mn alloys were tested and are displayed in Figure 9. The alloys show no damping peaks, except alloy II in the curves. When the temperature is lower than 300 °C, it has less effect on the damping capacities in alloy I and alloy III. When the temperature is higher than 300 °C, the values of internal friction (Q^−1^) grow exponentially. In alloy II, an internal friction peak appears in the range of 100 °C from 70 °C to 170 °C appears, which could be indexed as a dislocation damping peak [32,33,34]. The low height of the peak shows that only a small amount of movable dislocations was induced by Mn particles. At low working temperature (before 300 °C), it can be found that the Q^−1^ difference between the alloys is obvious, which shows that the damping value gradually decreases with the increase of Mn. The main reason is that the addition of Mn element promotes the morphological transformation of the LPSO phase and increase of dislocations. The high Mn-content alloy (alloy V) shows 0.0015 of Q^−1^ and only 16% of that of alloy II (0.009).

The temperature-dependent damping performance is also affected by many factors, such as the movement of dislocations, changes in morphology, and the increase of the second phase. Figure 10 shows the Q^−1^ values of the alloys at 250 °C and 400 °C. The change trend of alloy damping performance with Mn content at 250 °C is consistent with the test results at room temperature, both of which increase first and then decrease. The Q^−1^ at 400 °C shows that the change trend of the damping of the five alloys at high temperature is opposite to that at low temperature. The damping behavior at high temperature is very complicated, and it is related to grain boundary movement, dislocation slip or climbing, and phase transition.

### 3.3. Mechanical Properties

Figure 11 is the stress–strain curve of this experiments Mg-Zn-Y-Mn alloys. The mechanical performance of the alloys almost monotonously increases with the addition of Mn content, as shown in Table 3. The results indicate that the addition of a small amount Mn element cannot effectively improve the mechanical properties in the Mg-Zn-Y alloy. When the Mn content is more than 1.02%, a large number of Mn particles will be dispersed in the alloy matrix, and they have the effect of grain refinement. According to Figure 1 and Figure 2, it can also be clearly found that alloy V has the finest grain structure, so the mechanical performance of the alloys has been significantly improved. It can be explained by the Hall–Petch formula [35]:$$\sigma_{s}=\sigma_{0}+{\mathrm{Kd}}^{-\frac{1}{2}}$$

$\sigma_{s}$ represents yield strength, $\sigma_{0}$ represents the resistance to deformation inside the crystal, *d* represents the average grain size, and *K* represents the influence coefficient of grain boundary on deformation. Alloy V shows the best mechanical property in this experiment, its ultimate tensile strength (UTS) is 292 MPa, yield strength (YS) is 167 MPa, and elongation (δ) is 9.4%.

## 4. Conclusions

In this study, we design high performance Mg-Zn-Y-Mn series alloys with traditional casting. The microstructure, damping performance, and mechanical performance were discussed. The main conclusions were summed up, as follows:The addition of Mn can affect the morphology of LPSO second phase in Mg-Zn-Y alloy. The Mn element tends to diffuse to the LPSO interface, hinders the LPSO orientation growth, and causes the LPSO phase to expand in the arc direction. When the content of Mn is more than 1.02%, the morphology of LPSO phase shows a tendency of arc.The second phase morphology also has a significant effect on the damping performance of magnesium alloys. The circular structure of LPSO can improve the damping property of the alloy better than the structure with strong orientation, especially at the strain of 10^−3^ and 250 °C.While Mn is less than 0.44%, it has a positive influence on the damping behavior. With greater additions of Mn, damping values would have a reduction due to the dispersed fine LPSO phases and α-Mn particles. Alloy II shows the best damping property at room temperature.When the Mn content is higher than 1.02%, the Mn element can obviously refine the grain, and the mechanical performance of the alloys has been significantly improved. Alloy V shows the best mechanical property, its ultimate tensile strength (UTS) is 292 MPa, yield strength (YS) is 167 MPa, and elongation (δ) is 9.4%.

## Figures and Tables

**Figure 1 materials-13-04654-f001:**
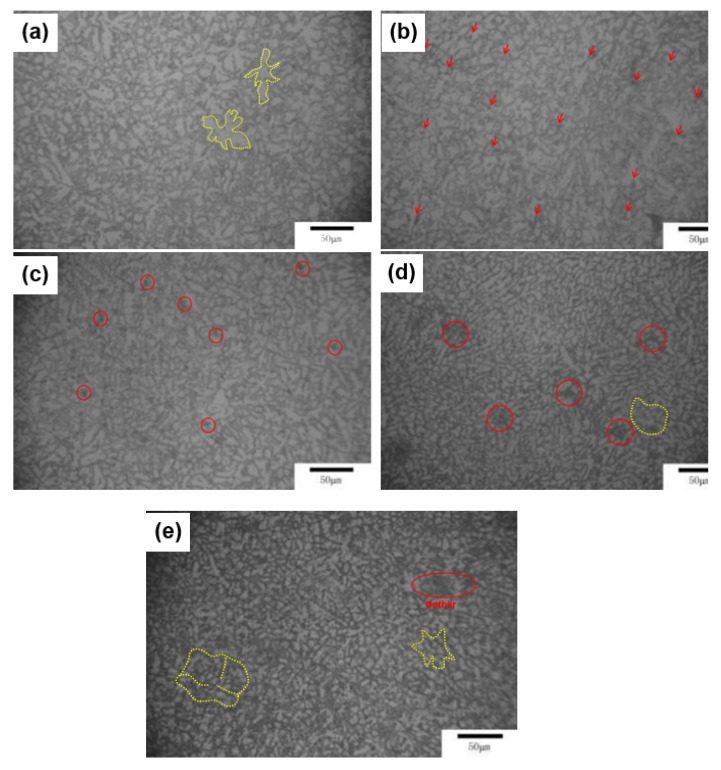
Optical micrographs of the as-cast Mg-Zn-Y-Mn alloys. (**a**) Alloy I; (**b**) Alloy II; (**c**) Alloy III; (**d**) Alloy IV; and, (**e**) Alloy V.

**Figure 2 materials-13-04654-f002:**
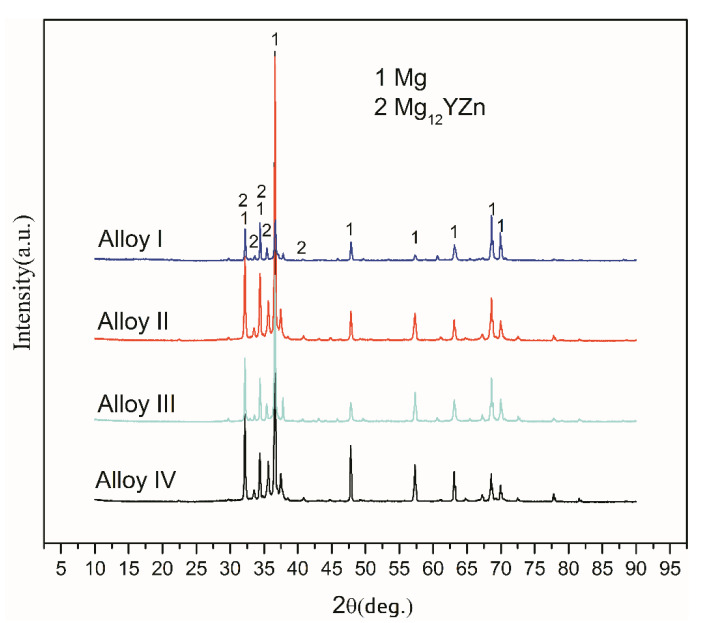
X-ray diffractometer (XRD) patterns of Mg-Zn-Y-Mn alloys.

**Figure 3 materials-13-04654-f003:**
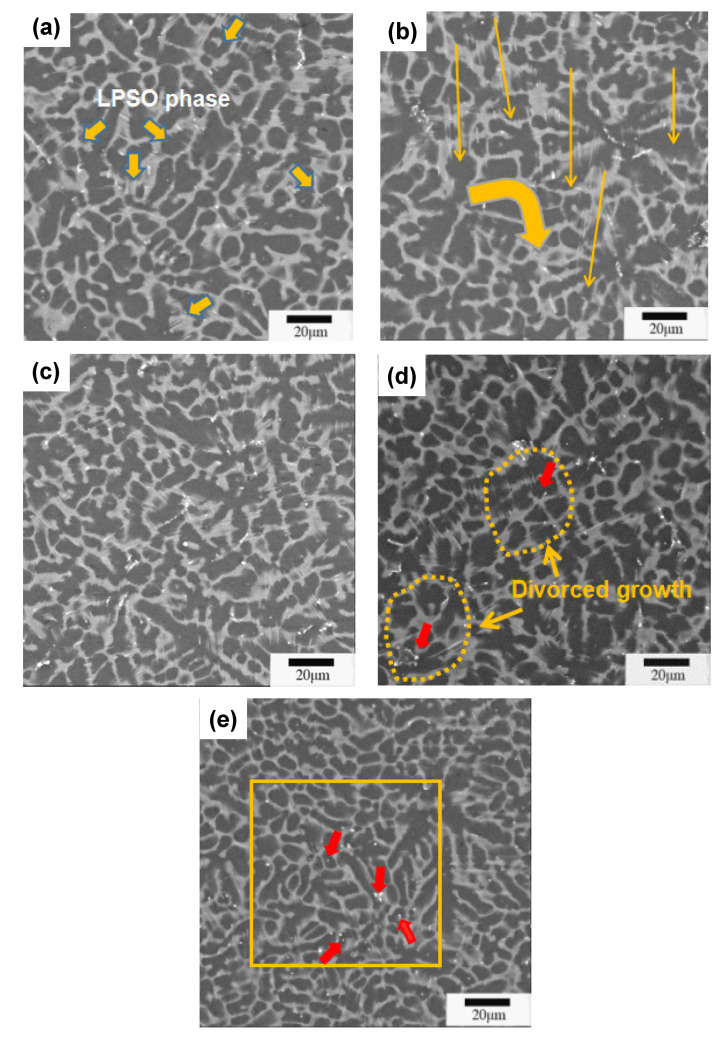
Scanning electron microscope (SEM) images of the as-cast Mg-Zn-Y-Mn alloys. (**a**) Alloy I; (**b**) Alloy II; (**c**) Alloy III; (**d**) Alloy IV; (**e**) Alloy V.

**Figure 4 materials-13-04654-f004:**
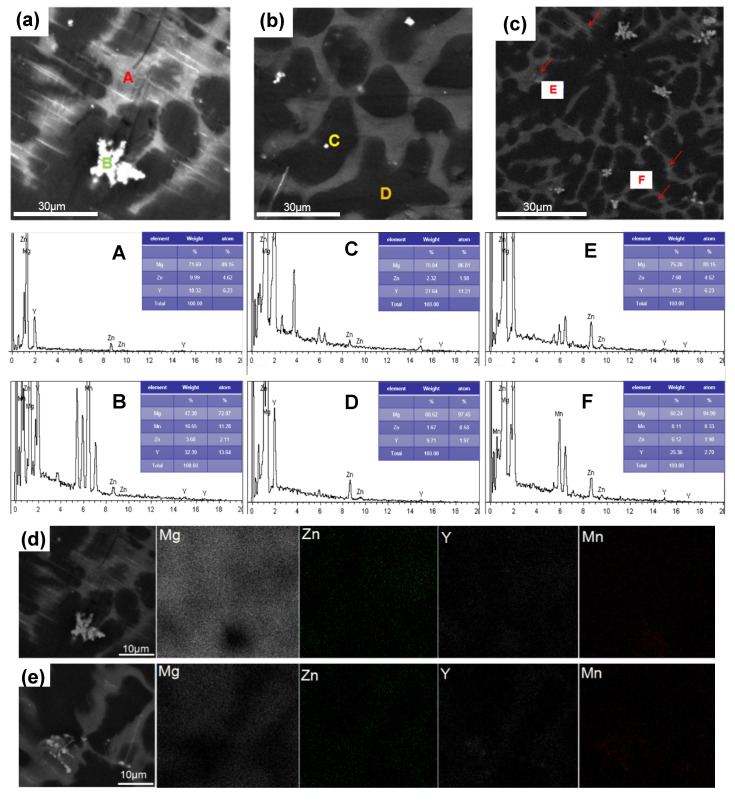
(**a**–**c**) SEM images of the as-cast Alloy V; (**A**–**F**) energy-dispersive spectrometer (EDS) element spectrum at different positions corresponding to Figure 4 (**a**–**c**), respectively; (**d**,**e**) Distribution of element of SEM microstructure in Alloy V, Alloy II respectively.

**Figure 5 materials-13-04654-f005:**
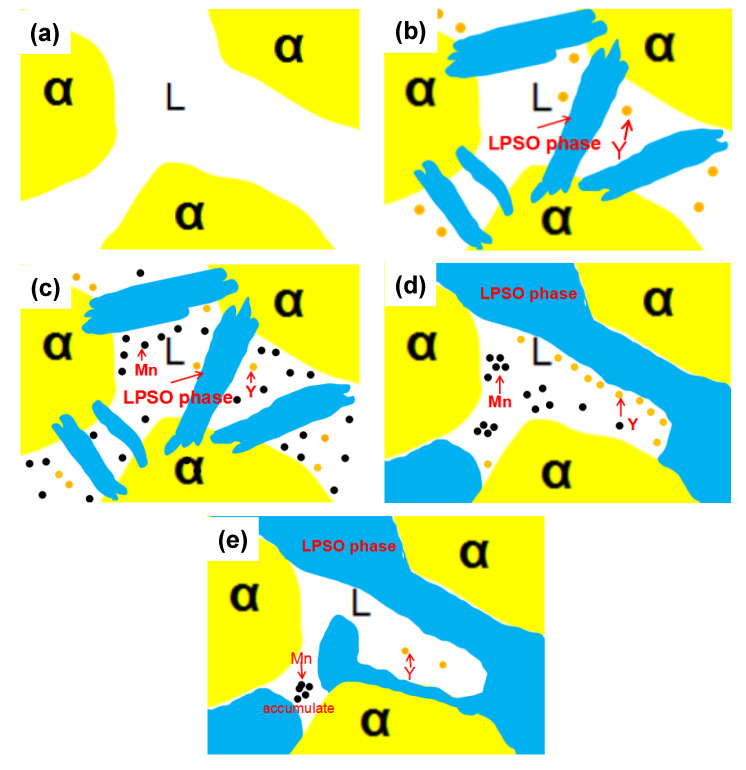
The diagram of divorced growth of Mg-4.9Zn-8.9Y series alloys. In the alloy without Mn, (**a**) α-Mg solidifies first, and then (**b**) precipitate LPSO phase and Y particles. After the addition of the Mn element, (**c**) the Mn element tends to diffuse to the LPSO interface at first; (**d**) Mn element will hinder the growth of LPSO; (**e**) The growth of LPSO drive Mn element to gather together.

**Figure 6 materials-13-04654-f006:**
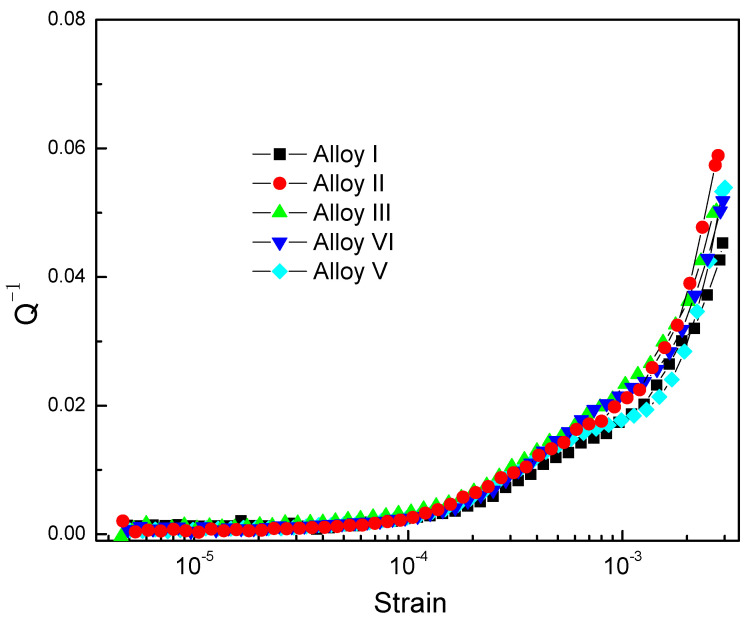
Damping capacities of as-cast Mg-Zn-Y-Mn alloys with different Mn contents.

**Figure 7 materials-13-04654-f007:**
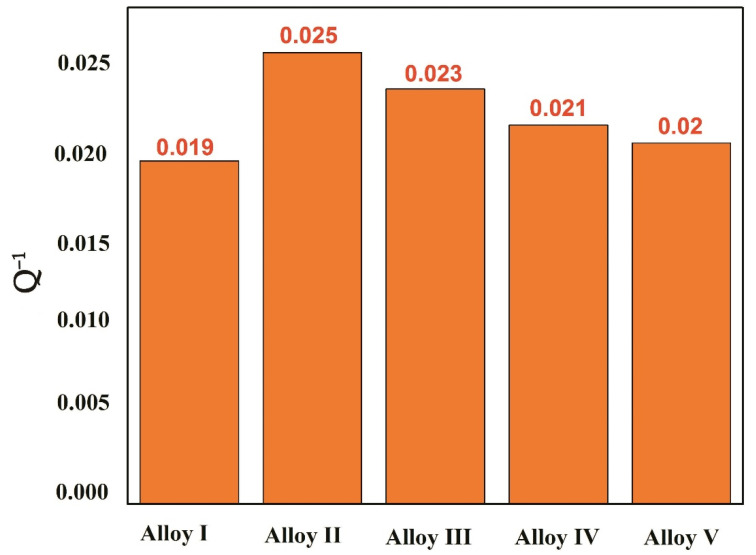
Damping performance of these five alloys at strain 10^−3^.

**Figure 8 materials-13-04654-f008:**
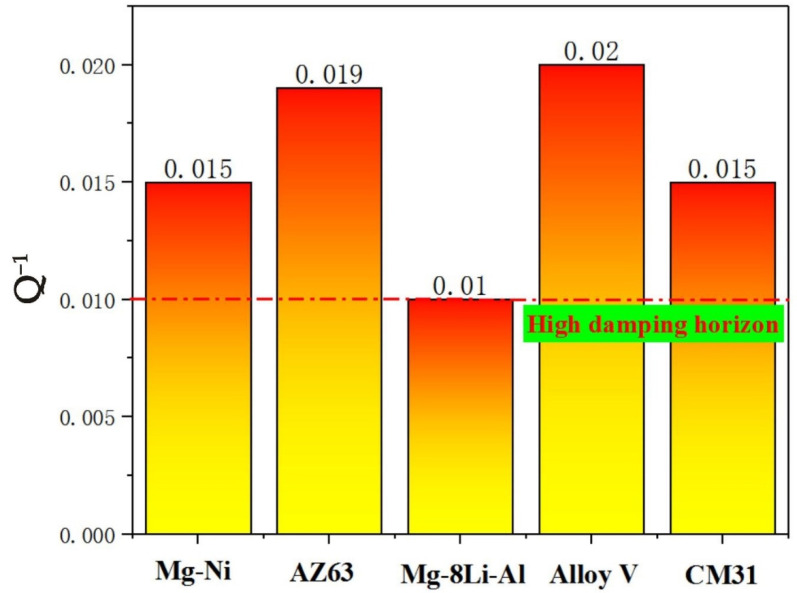
The comparison of the Q^−1^ between the Mg-4.9Zn-8.9Y-1.33Mn alloy V and Mg-Ni alloy [28], AZ63 alloy [29], Mg-8Li-Al alloy [30], and CM31 alloy [31].

**Figure 9 materials-13-04654-f009:**
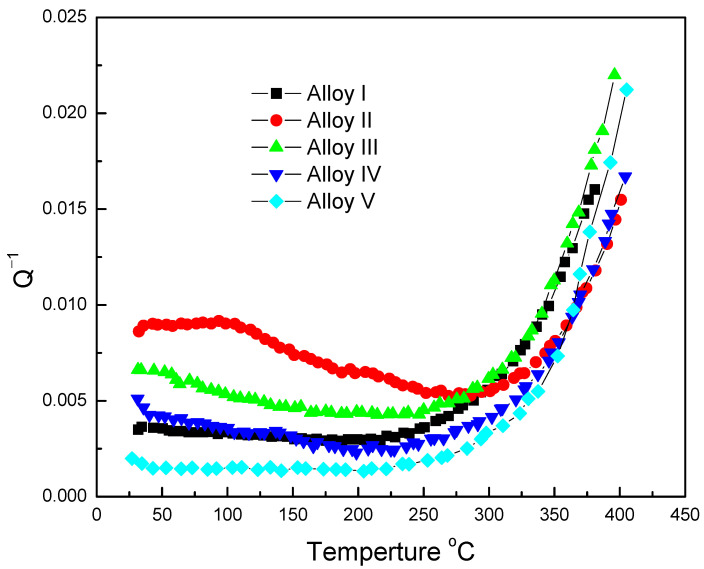
Relationship between damping capacities and temperature of Mg-Zn-Y-Mn alloys.

**Figure 10 materials-13-04654-f010:**
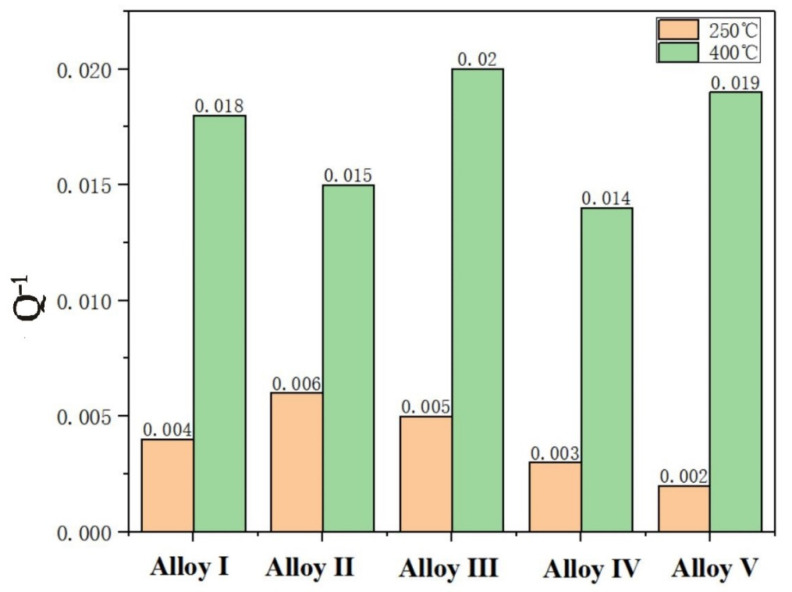
Damping performance of these five alloys at 250 ℃ and 400 ℃ respectively.

**Figure 11 materials-13-04654-f011:**
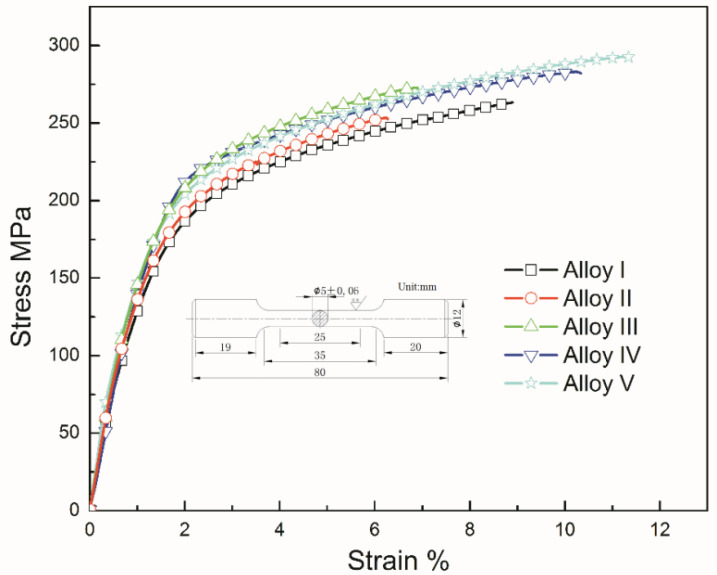
Tensile stress-strain curves of Mg-Zn-Y-Mn alloys.

**Table 1 materials-13-04654-t001:** The chemical composition Mg-Zn-Y-Mn alloys.

Alloy No.	Nominal Composition (wt.%)	Chemical Composition (wt.%)			
	Mg-4.9%Zn-8.9%Y-x%Mn	Mg	Zn	Y	Mn
Alloy I	x = 0	Bal.	4.42	9.50	-
Alloy II	x = 0.33	Bal.	4.36	9.58	0.44
Alloy III	x = 0.66	Bal.	4.99	9.54	0.64
Alloy IV	x = 1	Bal.	4.89	9.65	1.02
Alloy V	x = 1.33	Bal.	5.13	9.73	1.37

**Table 2 materials-13-04654-t002:** EDS elemental analysis of the as-cast Alloy V specified in Figure 4.

Point	Composition (at.%)			
	Mg	Zn	Y	Mn
A	89.34	4.39	6.27	0
B	77.45	1.70	5.03	15.82
C	86.81	1.98	11.21	0
D	97.45	0.58	1.97	0
E	89.15	4.62	6.23	0
F	94.99	1.98	2.70	0.33

**Table 3 materials-13-04654-t003:** Mechanical properties of Mg-Zn-Y-Mn alloys.

Alloy No.	Mechanical Properties		
	Ultimate Tensile Strength (MPa)	Yield Stress (MPa)	Elongation (%)
Alloy I	263	140	6.8
Alloy II	253	145	4.6
Alloy III	271	167	5.3
Alloy IV	282	158	8.5
Alloy V	292	167	9.4

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
