# Peer review of "Influence of Long-Period-Stacking Ordered Structure on the Damping Capacities and Mechanical Properties of Mg-Zn-Y-Mn As-Cast Alloys"

_materials, 2020, doi:10.3390/ma13204654_

Round 1

Reviewer 1 Report

     The authors investigated the microstructure, damping capacities and mechanical properties of Mg–Zn–Y–Mn alloys.

1. Figure1 (b) : Describe the meaning of the red arrows in the main text or caption of figure 1.
2. P8L166-167 : “The final structure of the second phase circular arc are expected to improve the damping and mechanical properties of the alloy.” Describe the reason of this sentence in the main text.
3. Figure 7 : Show each error (standard deviation) from Alloy I to V. In addition, explain there is the difference that each value is meaningful.
4. Table 3 : Write the official name of each abbreviation in the main text or caption of Table 3. For example, ultimate tensile strength (UTS), yield strength (YS), elongation (δ).

Author Response

1.Figure1 (b) : Describe the meaning of the red arrows in the main text or caption of figure 1.

The red arrow indicates that with the increase of Mn, some particles appear in alloy II in Figure 1b.The red circle in Figure 5c、5d represents Mn particles aggregated in alloy III to form a cluster structure.The red circle in Figure 5e represents further aggregation of Mn particles.

2.P8L166-167 : “The final structure of the second phase circular arc are expected to improve the damping and mechanical properties of the alloy.” Describe the reason of this sentence in the main text.

As the addition of Mn can affect the morphology of LPSO phase in as-cast Mg-Zn-Y alloy. The Mn element tends to diffuse to the LPSO interface, hinders the LPSO orientation growth, and causes the LPSO phase to expand in the arc direction. When the content of Mn is more than 1.02%, the morphology of LPSO phase shows a tendency of arc. the Mn element can effectively refine the grain, and the mechanical properties of the alloys have been significantly improved. Alloy V shows the best mechanical property, its ultimate tensile strength is 292 MPa, yield strength is 167 MPa and elongation is 9.4%.

  1. Figure 7 : Show each error (standard deviation) from Alloy I to V. In addition, explain there is the difference that each value is meaningful.

The difference in damping at room temperature is not so obvious because the alloying elements are not much different, but it is more obvious in the heating damping. Therefore, for the room temperature damping, only the damping at the intercept and high strain is used for qualitative analysis and comparison.

4.Table 3 : Write the official name of each abbreviation in the main text or caption of Table 3. For example, ultimate tensile strength (UTS), yield strength (YS), elongation (δ).

The corresponding part has been changed in the revised manuscript.

Table 3. Mechanical properties of Mg-Zn-Y-Mn alloys.

Alloy No.

Mechanical properties

Ultimate Tensile Strength (MPa)

Yield Stress(MPa)

Elongation(%)

Alloy I

263

140

6.8

Alloy II

253

145

4.6

Alloy III

271

167

5.3

Alloy IV

282

158

8.5

Alloy V

292

167

9.4

Reviewer 2 Report

The paper is topical. Results are original and are worthy publishing in Materials. The paper must be however completely rewritten for the English style and grammar.

A few examples are:

P1. Line 20  “ Alloy V shows the best mechanical property»  Please avoid using undefined designations in the Abstract. What is alloy V?  Do you mean one mechanical properties or a combination of mechanical properties here?

P.2 Line 35. «However the mechanical properties and damping 35 properties of alloys have always been contradictory.» Please reword, unclear sentence.

P2. Line 38 Incomplete sentence.

P2. Line 41 «  As the most ternary 40 alloys LPSO alloys... » Wrong English. Reword.

Errors in English are too numerous to be listed here. It is the responsibility of the authors to deliver their message clearly. The authors fail to express themselves in a scientifically sound manner. The authors are encouraged to proofread their manuscript with the English first editor. The present reviewer will not accept any superficial revision.

Page 2 Line 75. Define Q^-1.

Section 3.1. Provide the estimates for the volume fraction of the LPSO phase in the alloys studied.

line 138 – What do you mean by “divorced” here? Please reword.

Section 3.2.

Fig. 7 Provide the confidence bands for Q^-1 for each alloy. How many specimens of each kind have been tested?  The data shown in Fig. 6 and 7 does not convince me that the damping capacity is different for different alloys studied (cf. authors statement in line  180). Please show the results of independent testing of several specimens of the same kind for reproducibility  of results.  

Line 171. I do not see any “critical” strain value delineating different damping regimes. The authors conclusion “the critical strain could be indexed as 10-4» is fully artificial and is not supported by any arguments.  

Fig. 8, do the authors compare the Q^-1 values obtained under comparable conditions for the alloys shown? Provide the confidence bands! I do not see the grounds for such a comparison.

Table 3. Provide the confidence intervals for the quantities shown. How many tensile tests have been performed for the alloys of each kind?  Is the yield stress is measured as the 0.2% proof stress?

Author Response

1.P1. Line 20  “ Alloy V shows the best mechanical property»  Please avoid using undefined designations in the Abstract. What is alloy V?  Do you mean one mechanical properties or a combination of mechanical properties here?

(1)Alloy V represents Mg-4.9%Zn-8.9%Y-1.33%Mn.

(2)Mechanical properties represents its ultimate tensile strength is 292 MPa, yield strength is 167 MPa and elongation is 9.4%.

2.P.2 Line 35. «However the mechanical properties and damping 35 properties of alloys have always been contradictory.» Please reword, unclear sentence.

Corrected to«However, materials with high damping capacities generally exhibit poor mechanical properties.»

3.P2. Line 38 Incomplete sentence.

Corrected to«The relationship between equilibrium mechanics and damping has always been a research hotspot.»

4.P2. Line 41 «  As the most ternary 40 alloys LPSO alloys... » Wrong English. Reword.

Corrected to«As the most common ternary alloy in alloys containing LPSO phase.»

5.Page 2 Line 75. Define Q^-1.

Q-1 is the inverse of the quality factor, used to measure the capacities of the material damping.

6.Section 3.1. Provide the estimates for the volume fraction of the LPSO phase in the alloys studied.

The volume fraction of LPSO phase in the five alloys is not significantly different, about 38%.

7.line 138 – What do you mean by “divorced” here? Please reword.

Divorced growth is explained as the addition of Mn can affect the morphology of LPSO phase in as-cast Mg-Zn-Y alloy. The Mn element tends to diffuse to the LPSO interface, hinders the LPSO orientation growth, and causes the LPSO phase to expand in the arc direction. When the content of Mn is more than 1.02%, the morphology of LPSO phase shows a tendency of arc. 

8.Fig. 7 Provide the confidence bands for Q^-1 for each alloy. How many specimens of each kind have been tested?  The data shown in Fig. 6 and 7 does not convince me that the damping capacity is different for different alloys studied (cf. authors statement in line  180). Please show the results of independent testing of several specimens of the same kind for reproducibility of results.  

The damping test result here is obtained by taking three averages for each sample.

As you said, the difference in damping at room temperature is not so obvious because the alloying elements are not much different, but it is more obvious in the heating damping. Therefore, for the room temperature damping, only the damping at the intercept and high strain is used for qualitative analysis and comparison.

9.Line 171. I do not see any “critical” strain value delineating different damping regimes. The authors conclusion “the critical strain could be indexed as 10-4» is fully artificial and is not supported by any arguments. 

The damping of magnesium alloys is generally regarded as a dislocation-type damping mechanism, which conforms to the G-L theory. Therefore, the damping of magnesium alloys can be divided into two stages: when the strain is low, the damping has nothing to do with the strain, and the mechanical energy will be consumed by the bowing motion of the dislocation line in the alloy. When the strain exceeds the critical value, the dislocation line drags the weak pinning point to move together, and the damping of the alloy is greatly increased at this time. For complex systems, the critical unpinning strain is often not obvious. In this experiment, it can be observed that when the strain is 10-4~2*10-4, the damping value begins to increase significantly.

  1. 8, do the authors compare the Q^-1 values obtained under comparable conditions for the alloys shown? Provide the confidence bands! I do not see the grounds for such a comparison.

Sorry, it is difficult for us to do this. The data in Figure 8 are all obtained from other documents. We cannot obtain their complete data for comprehensive comparison. Since the damping of magnesium alloy is mainly affected by the strain amplitude, we Only the damping value under representative strain can be selected for qualitative evaluation.

If Figure 8 is considered meaningless, we also consider deleting it.

  1. Table 3. Provide the confidence intervals for the quantities shown. How many tensile tests have been performed for the alloys of each kind?  Is the yield stress is measured as the 0.2% proof stress?

Take three samples for each mechanical experiment and take the average value. 

Reviewer 3 Report

1)The microstructure (lines 106-167) should be described better (explanation)

2) The obtained results are very important. However their interpretation is missing.

3) Shape, dimensions of tensile specimens should be given

Author Response

  1. The microstructure (lines 106-167) should be described better (explanation).

We have revised this section to make it clearer.

  1. The obtained results are very important. However their interpretation is missing.

(1)When the Mn content is higher than 1.02%, the Mn element can effectively refine the grains, and the grain diameter d decreases, which can be explained by the Hall-Petch formula:

The mechanical properties of the alloy have been significantly improved. Alloy V shows the best mechanical properties, with its ultimate tensile strength of 292MPa, yield strength of 167MPa, and elongation of 9.4%.

(2)The second phase morphology also has a significant effect on the damping of magnesium alloys. The circular structure of LPSO can improve the damping property of the alloy better than the structure with strong orientation, especially at the strain of 10-3 and 250 ℃.

With the increase of Mn content, the orientation of the LPSO phase is weakened, and bent knots appear. Each bent knot interacts with the surrounding Y and Mn particles to drive the solute atoms to move together. When the solute atoms change their positions through diffusion movement, it is necessary For a certain period of time, internal friction occurs, so the damping performance of alloy II and III is relatively excellent.When the Mn content increases to 1.37%, some Mn atoms gather in the grain boundaries, and do not act as pinning points to hinder the movement of dislocations and the interaction with the LPSO phase. Only Y atoms are wrapped in the LPSO phase, and the energy dissipation is weakened. The damping performance has been reduced. In general, the damping performance is still improved compared to Alloy I without Mn added.

The addition of a small amount of Mn causes a large number of dislocations around the Mn to appear due to the different thermal expansion coefficients during the alloy cooling process, which provides movable dislocations for the alloy, which is beneficial to improve the damping of the alloy. When the Mn content is high, the dislocations will hinder each other, so it is no longer beneficial to the damping of the alloy.

3.Shape, dimensions of tensile specimens should be given.

Shape and size of rod-shaped tensile specimen

Round 2

Reviewer 2 Report

The content of the paper has improved. Even though the experimental results might be interesting, the presentation is still below any critics.  The authors have not proofread their carelessly written first submission and the quality of the English is still so poor that it the paper is practically not readable.  The newly inserted text also suffers from multiple awkward and wrong expressions. For example,  p.4 lines 114-127 - errors are in every sentence! E.g.  "Fig. 3(b) shows that the orientation of the LPSO Structure growth begins to weaken" is impossible to understand!

p. 8 Line 177 .  "G-L theory". I can suppose the authors refer to the "Granato and Lucke theory"  

It is the responsibility of the authors to ensure the quality of the language, and I do not recommend the paper for publication until they improve it significantly. 

Author Response

1.The content of the paper has improved. Even though the experimental results might be interesting, the presentation is still below any critics.  The authors have not proofread their carelessly written first submission and the quality of the English is still so poor that it the paper is practically not readable.  The newly inserted text also suffers from multiple awkward and wrong expressions. For example,  p.4 lines 114-127 - errors are in every sentence! E.g.  "Fig. 3(b) shows that the orientation of the LPSO Structure growth begins to weaken" is impossible to understand!

We have revised the grammar and expression problems again, and the revised parts are highlighted in red in the text(For examples,p.4 lines 114-127).

Fig. 3 shows the SEM images of the alloys, it can clearly see the dendrite structure and the second phase. The secondary phase presents a continuously distributed gray body on the grain boundary and forms a network structure. Some bright white particles are dispersed in the matrix. Fig.3a shows that the second phase of the alloy without Mn has an obvious orientation, the layered LPSO in each grain is parallel to each other. Fig. 3b,Fig. 3c shows that with the addition of Mn, the orientation of LPSO arranged at the grain boundary decreases. Interestingly, the main LPSO phases do not grow parallel to each other, and their morphology expands toward the arc of the yellow arrow. In Fig. 3d, the preferential growth of LPSO is no longer obvious, and secondary phase expands and connects to form a network along the arc direction, as shown in the yellow dotted line. This phenomenon is consistent with the speculated results of the metallographic observation. While Mn reached 1.37% in alloy V, it is observed that bright white dots are surrounded by LPSO phase in Fig. 3e. With the addition of Mn, the as-cast morphology of the alloy changed slightly.

2. 8 Line 177 .  "G-L theory". I can suppose the authors refer to the "Granato and Lucke theory"  

The reference to G-L theory has been re-added the reference:

  • Granato, K. Lüker. Theroy of Mechanical Damping Due to Dislocation.J. Appl. Phys. 1956, 27,6, 583-593.[CrossRef]
  • Granato, K. Lüker. Application of Dislocation Theory to Internal Friction Phenomena at High Frequencies. J. Appl.Phys. 1956, 27, 7, 789-805.[CrossRef]

This manuscript is a resubmission of an earlier submission. The following is a list of the peer review reports and author responses from that submission.